# Genetic Basis of Seedling Root Traits in Common Wheat (*Triticum aestivum* L.) Identified by Genome-Wide Linkage Mapping

**DOI:** 10.3390/plants14030490

**Published:** 2025-02-06

**Authors:** Xiaole Ma, Juncheng Wang, Hong Zhang, Lirong Yao, Erjing Si, Baochun Li, Yaxiong Meng, Huajun Wang

**Affiliations:** 1College of Agronomy, Gansu Agricultural University, Lanzhou 730070, China; maxl@gsau.edu.cn (X.M.); wangjc@gsau.edu.cn (J.W.); zhangh@gsau.edu.cn (H.Z.); yaolr@gsau.edu.cn (L.Y.); siej@gsau.edu.cn (E.S.); mengyx@gsau.edu.cn (Y.M.); 2State Key Laboratory of Aridland Crop Science/Gansu Key Laboratory of Crop Improvement and Germplasm Enhancement, Lanzhou 730070, China; libc@gsau.edu.cn

**Keywords:** common wheat (*Triticum aestivum* L.), linkage mapping, kompetitive allele-specific PCR (KASP), marker-assisted selection (MAS), quantitative trait loci (QTLs)

## Abstract

Common wheat production is significantly influenced by abiotic stresses. Identifying the genetic loci for seedling root traits and developing the available molecular markers are crucial for breeding high yielding and stable varieties. In this study, five wheat seedling root traits, including root length (RL), root surface area (RA), root volume (RV), number of root tips (RT), and root dry weight (RW), were measured in the Wp-072/Wp-119 recombinant inbred line (RIL) population. Genotyping was conducted for the RIL population and their parents using the wheat 90K single-nucleotide polymorphism (SNP) chip. In total, three quantitative trait loci (QTLs) for RL (*QRL.gau-1DS*, *QRL.gau-1DL* and *QRL.gau-4AL*), two QTLs for RA (*QRA.gau-1D* and *QRA.gau-2DL*), one locus for RV (*QRV.gau-6AS*), two loci for RW (*QRW.gau-2DL* and *QRW.gau-2AS*), and two loci for RT (*QRT.gau-3AS* and *QRT.gau-6DL*) were identified, with each explaining 4.5–8.4% of the phenotypic variances, respectively. Among these, *QRT.gau-3AS*, *QRL.gau-4AL*, and *QRV.gau-6AS* overlapped with the previous reports, whereas the other seven QTLs were novel. The favorable alleles of *QRL.gau-1DS*, *QRL.gau-1DL*, *QRL.gau-4AL*, *QRA.gau-1D*, *QRW.gau-2AS*, *QRV.gau-6AS*, *QRT.gau-3AS*, and *QRT.gau-6DL* were contributed by Wp-072, whereas the other two loci originated from Wp-119. Additionally, five kompetitive allele-specific PCR (KASP) markers, *KASP-RL-1DL* for RL, *KASP-RA-1D* and *KASP-RA-2DL* for RA, *KASP-RW-2AS* and *KASP-RW-2DL* for RW, were developed and validated successfully in 149 wheat accessions. Furthermore, seven candidate genes mainly for plant hormones were selected and validated by quantitative real-time PCR (qRT-PCR). This study provides new loci, new candidate genes, available KASP markers, and varieties for optimizing wheat root system architecture.

## 1. Introduction

Common wheat production is significantly influenced by abiotic stresses [1,2]. In recent years, climate change has intensified the effects of stress factors on wheat production, leading to yield losses of 35–50% (https://www.natesc.org.cn/, accessed on 19 January 2025). To actively address the significant losses caused by abiotic stresses such as drought, salinity, and alkalinity, it is imperative to harness modern biotechnologies to enhance crop tolerance and ensure the security of wheat production [3,4,5,6]. The root system plays a pivotal role in anchoring a plant, absorbing and transporting water and nutrients while also storing essential substances [3,5]. A well-developed root architecture serves as the genetic basis for the water-saving, salt-tolerance, and lodging-tolerance traits in wheat [7]. In the process of improving wheat landraces into cultivars, optimizing the root system has played a pivotal role. This optimization has not only significantly boosted yield and stress tolerance but has also underscored the importance of favorable root architecture [6,7,8].

Wheat is a typical fibrous-rooted crop. Adventitious roots, which form during post-embryonic development, are the core tissues responsible for plant anchorage and the absorption of water and nutrients [3,4]. The composition of adventitious roots during the seedling stage is a crucial factor in root system architecture. The development of the root system at the adult stage is easily influenced by soil conditions and cultivation practices, and conventional sampling methods are highly destructive and complex, making phenotypic investigations challenging and inefficient [5,6]. Previous studies have shown that root system heritability is high during the seedling stage, with minimal environmental influence, and can reflect the root architecture and distribution at the adult stage. This characteristic is closely related to abiotic tolerance and yield [7]. The root length, root surface area, root volume, and the number of root tips influence the spatial arrangement of the root system, significantly affecting water and nutrient uptake [9]. The formation of adventitious roots in wheat is influenced by both internal and external factors. External factors primarily include light, carbon sources, nitrogen sources, inorganic salts, metal ions, and trace elements. Among these, light (photoperiod, light quality, light intensity, and light cycle) is the most significant external factor affecting the production of adventitious roots. Plant hormones are the key internal factors influencing root development. The primary regulatory hormones are auxins (IAAs), followed by cytokinins (CTKs) and ethylene (ETH) [10,11].

Currently, major wheat-producing countries, including the United States, Canada, Australia, and the International Maize and Wheat Improvement Center (CIMMYT) have identified root system optimization as a key breeding objective [9]. However, the measurement of adventitious roots in wheat is time-consuming, labor-intensive, and often damaging to the plants, making it impractical for breeders to select for them directly in the field. Marker-assisted selection (MAS) has thus become an effective method for optimizing the root system [12,13]. Identifying the loci responsible for regulating wheat seedling root traits, cloning target genes, validating gene functions, and developing gene-specific molecular markers are the foundations of MAS breeding [6,9,14]. To improve wheat seedling root traits, it is essential to identify the significant genomic regions and develop practical molecular markers [8. Previous studies have reported that wheat seedling root traits are governed by multiple minor genes [8,15,16]. Linkage and association mapping have been widely employed to elucidate the genetic basis of complex traits in wheat, leveraging advancements in high-throughput genotyping technologies [17]. To date, numerous quantitative trait loci (QTLs) have been identified for wheat seedling root traits, primarily distributed on chromosomes 1A, 1D, 2A, 3A, 3B, 4B, 5A, 5B, 5D, 6A, and 7B [5,8,16,18,19,20,21,22,23].

Although several studies on seedling root traits at the seedling stage have been conducted, the identified genetic regions are broad, and the markers are not tightly linked, rendering them ineffective for practical breeding applications. Therefore, it is crucial to elucidate the genetic mechanisms of seedling root traits, develop tightly linked markers, and conduct gene pyramiding during breeding. In this study, we conducted linkage mapping for seedling root traits using the wheat 90K SNP array in a bi-parental recombinant inbred line (RIL) population derived from a Wp-072/Wp-119 cross. The objective of this study is to uncover the genetic basis of wheat seedling root traits, leveraging the available kompetitive allele-specific PCR (KASP) markers and the outstanding wheat accessions to improve seedling root traits.

## 2. Results

### 2.1. Phenotypic Evaluation of Wheat Seedling Root Traits

A total of five seedling root traits showed continuous and significant variations across 243 RILs (Figure 1). Wp-072 is an advanced breeding line with a higher tolerance to abiotic stress and with well-developed root systems. In contrast, Wp-119 has poor abiotic stress tolerance and weaker root development. The mean values for RL, RA, RV, RW, and RT of Wp-072 and Wp-119 were 79.5 cm and 65.2 cm, 10.6 mm^2^ and 7.9 mm^2^, 442.3 mm^3^ and 289.6 mm^3^, 0.033 g and 0.026g, 236.1 and 203.2, respectively (Figure 1). The mean values for RL, RA, RV, RW, and RT of the RIL population were 73.5 cm (range: 24.8–146.3 cm), 10.3 mm^2^ (4.0–17.5 mm^2^), 370.1 mm^3^ (22.0–1027.0 mm^3^), 0.030 g (0.015–0.046 g), and 214.6 (57.8–389.3), respectively (Appendix A). The standard deviations (stds) and coefficients of variation for root length (RL), root surface area (RA), root volume (RV), root dry weight (RW), and number of root tips (RT) were 28.1 cm (38.0%), 2.7 mm^2^ (26.0%), 208.2 mm^3^ (56.0%), 0.006 g (19.0%), and 64.7 (30.0%), respectively. Significant correlations were observed between RL, RA, and RT, with correlation coefficients of 0.713 (*p* < 0.01) between RL and RA, 0.256 (*p* < 0.05) between RL and RT, and 0.273 (*p* < 0.05) between RL and RV. Similarly, significant correlations were observed among RA, RW, and RT, with correlation coefficients of 0.235 (*p* < 0.05) between RA and RW, as well as 0.252 (*p* < 0.05) between RA and RT. Additionally, RV was significantly associated with RT (*R*^2^ = 0.678, *p* < 0.01), and RW was significantly associated with RT (*R*^2^ = 0.251, *p* < 0.05) (Appendix A).

### 2.2. Construction of the Linkage Map

Using the wheat 90K SNP chip, we genotyped the Wp-072/Wp-119 RIL population and constructed a high-density genetic linkage map. This linkage map contains 2243 backbone markers representing 6525 SNPs. The map length is 2924.4 cM, with an average length of 138.9 cM per chromosome and an average marker spacing of 1.30 cM. Among the 21 chromosomes, 1B, 2B, and 6B have the most markers, while 4D and 6D have the fewest markers. The B genome contains the most markers (41.0%), followed by the A genome (44.8%), while the D genome contains the fewest markers (14.2%).

### 2.3. QTL Detection for Seedling Root Traits

Three loci for RL were detected on chromosomes 1D and 4A. These loci are referred to as *QRL.gau-1DS* (7.1–7.9 Mb, flanked by *Excalibur_c17152_454* and *RAC875_c1471_566*), *QRL.gau-1DL* (424.7–431.8 Mb, flanked by *RAC875_c103613_441* and *Excalibur_c1236_840*), and *QRL.gau-4AL* (731.4–732.5 Mb, flanked by *Kukri_c18350_151* and *wsnp_Ex_rep_c70574_69491038*). These loci explained 5.2% (additive effect (add): −6.1), 8.4% (add: −9.2), and 5.4% (add: −6.1) of the total PVEs, respectively (Table 1). All favorable alleles for *QRL.gau-1DS*, *QRL.gau-1DL*, and *QRL.gau-4AL* were contributed by Wp-072.

For RA, the following two QTLs were identified: *QRA.gau-1D* (294.5–302.8 Mb, flanked by *BobWhite_c6770_617* and *BS00026262_51*) and *QRA.gau-2DL* (439.6–446.0 Mb, flanked by *RAC875_c55313_89* and *Kukri_c64788_552*). These QTLs explained 8.2% (add: -0.7) and 8.4% (add: 0.7) of the total PVEs, respectively. The favorable allele of *QRA.gau-1D* was contributed by Wp-072, while the favorable allele of *QRA.gau-2DL* was contributed by Wp-119. *QRV.gau-6AS* was located on the genetic interval of 31.0–45.3 Mb on chromosome 6AS and flanked by *BS00059454_51* and *Kukri_c19883_816*, explaining 5.1% (add: −58.2) of the PVEs. The favorable allele of *QRV.gau-6AS* was contributed by Wp-072.

The following two QTLs for RW were located on chromosomes 2A and 2D: *QRW.gau-2AS* (203.0–208.5 Mb, flanked by *Ex_c31468_763* and *tplb0055d02_624*) and *QRW.gau-2DL* (623.3–629.2 Mb, flanked by *CAP11_c4727_205* and *Tdurum_contig11539_81*). These QTLs explained 8.2% (add: 0.0186 g) and 8.1% (add: 0.0182 g) of the total PVEs, respectively. The favorable alleles of *QRW.gau-2AS* and *QRW.gau-2DL* were contributed by Wp-072 and Wp-119, respectively. For RT, the following two QTLs were identified: *QRT.gau-3AS* (50.0–70.2 Mb, flanked by *BS00039489_51* and *Kukri_c15325_1360*) and *QRT.gau-6DL* (462.6–468.3 Mb, flanked by *RAC875_c66820_684* and *Kukri_c22718_1072*). These QTLs explained 4.5% (add: −6.0) and 7.2% (add: −7.5) of the PVEs, respectively. All favorable alleles for *QRT.gau-3AS* and *QRT.gau-6DL* were contributed by Wp-072 (Table 1, Figure 2).

### 2.4. KASP Markers Development and Validation

Five QTLs with PVE > 8.0% were used to develop the KASP markers. Over 20 SNPs were tested for conversion into KASP markers in the Wp-072/Wp-199 RIL population. In addition, a total of 149 wheat varieties were used to validate the effectiveness of the KASP markers. Consequently, five KASP markers were successfully developed and validated in the natural population, including *KASP-RL-1DL* (*QRL.gau-1DL*, converted by *RAC875_c103613_441*, 431.8 Mb), *KASP-RA-1D* (*QRA.gau-1D*, corresponding to *wsnp_Ex_rep_c70574_69491038*, 302.8 Mb), *KASP-RA-2DL* (*QRA.gau-2DL*, originating from *BobWhite_c6770_617*, 439.6 Mb), *KASP-RW-2AS* (*QRW.gau-2AS*, converted by *BS00039489_51*, 203.0 Mb), and *KASP-RW-2DL* (*QRW.gau-2DL*, converted by *Kukri_c22718_1072*, 629.2 Mb) (Table 2, Figure 3).

For *KASP-RL-1DL*, the favorable allele (TT, 37.6%) showed a longer RL (88.6 cm) compared to the unfavorable allele (CC, 51.7%), which had a mean root length of 81.7 cm (*p* < 0.05) in the natural population. In the case of *KASP-RA-1D*, the favorable allele AA (37.6%) exhibited a higher mean RA of 12.5 mm^2^, as opposed to the unfavorable allele CC (56.4%) with a mean root area of 11.2 mm^2^ (*p* < 0.05) in the natural population. With *KASP-RA-2DL*, the favorable allele GG (35.6%, mean root area: 12.1 mm^2^) demonstrated a greater RA than the unfavorable allele AA (59.7%, mean root area: 10.7 mm^2^) (*p* < 0.05) in the natural population. For *KASP-RW-2AS*, the favorable allele AA (44.3%, mean root weight: 0.037 g) resulted in higher RW compared to the unfavorable allele GG (53.0%, mean root weight: 0.032 g), with the difference being statistically significant at the *p* < 0.05 level in the natural population. Similarly, for *KASP-RW-2DL*, the favorable allele AA (27.5%, mean root weight: 0.037 g) led to higher RW than the unfavorable allele GG (68.5%, mean root weight: 0.033 g) (*p* < 0.05) (Table 3 and Appendix A) in the natural population.

### 2.5. Candidate Gene Identification

Identifying the candidate genes for important target traits is crucial for conducting further fine mapping and gene cloning. A total of nine candidate genes, selected through QTL mapping and annotation, were found to be involved in the biological metabolism of plant hormones, leucine-rich repeat receptor-like protein kinase (LRR-RLK), and zinc finger family proteins (Table 4 and Appendix A). Specifically, the gene *TraesCS1D01G018200* from *QRL.gau-1DS* encodes a zinc finger family protein. Meanwhile, *TraesCS3A01G101400* from *QRT.gau-3AS* encodes an E3 ubiquitin protein ligase. The gene *TraesCS6A01G070300* from *QRV.gau-6AS* codes for an F-box family protein. Both *TraesCS2A01G220500* from *QRW.gau-2AS* and *TraesCS2D01G344600* from *QRA.gau-2DL* are involved in encoding LRR-RLKs. Four candidate genes related to plant hormones were identified. Of these, *TraesCS1D01G216500* from *QRA.gau-1D* corresponds to an auxin canalization protein. *TraesCS1D01G336900* from *QRL.gau-1DL* encodes gibberellin 2-oxidase. *TraesCS2D01G548900* from *QRW.gau-2DL* is an auxin response factor (ARF) gene. Lastly, *TraesCS6D01G383600* from *QRT.gau-6DL* codes for an ethylene (ETH) receptor. The expression levels of the nine candidate genes in the seedling roots of Wp-072 and Wp-119 were detected using quantitative real-time PCR (qRT-PCR). Among these, *TraesCS2A01G220500* and *TraesCS3A01G101400* showed no significant differences between Wp-072 and Wp-119, while *TraesCS1D01G018200*, *TraesCS1D01G216500*, *TraesCS1D01G336900*, *TraesCS2D01G344600*, *TraesCS2D01G548900*, *TraesCS6A01G070300*, and *TraesCS6D01G383600* exhibited a 2.3–8.0-fold higher expression in Wp-072 compared to Wp-119 (Figure 4).

## 3. Discussion

For a long time, wheat breeding efforts have primarily focused on above-ground traits. However, the research and improvement in seedling root traits have been significantly constrained due to the complexity of phenotypic evaluation [7]. A deeper understanding of the genetic basis underlying root system traits, coupled with the identification of novel genes and the development of KASP markers, will contribute to the enhancement of wheat root systems and lead to the high and stable yield of these crops. In this study, we identified three QTLs for root length (*QRL.gau-1DS*, *QRL.gau-1DL*, and *QRL.gau-4AL*), two loci for total root surface area (*QRA.gau-1D* and *QRA.gau-2DL*), a QTL for total root volume (*QRV.gau-6AS*), two QTLs for root dry weight (*QRW.gau-2DL* and *QRW.gau-2AS*), and two loci for number of root tips (*QRT.gau-3AS* and *QRT.gau-6DL*).

### 3.1. Seven Novel Loci for Wheat Root System Traits Were Identified

Although the research uncovering the genetic basis of wheat root system traits is limited, some studies have reported relevant findings. Jin et al. [14] identified a QTL on chromosome 4A for root length (*QTRL.caas-4A.2*, 732.6 Mb) in the Doumai/Shi4185 RIL population, which overlapped with *QRL.gau-4AL* (731.4–732.5 Mb) also identified in this study. Saini et al. [24] conducted a meta-analysis for wheat seedling root traits and identified six meta-QTLs on chromosome 4A. Among these, the *MQTL4A.1* (651.78–705.73 Mb) for root length, root volume, and number of root tips was nearly overlapping with *QRL.gau-4AL* (731.4–732.5 Mb) identified in this study. Sallam et al. [25] reported 11 loci for wheat seedling root traits on chromosomes 1A, 2A, 2B, 3A, 5B, 7A, and 7B. The locus tightly linked with *CAP8_c359_95* (74.4 Mb on chromosome 3A) was close to *QRT.gau-3AS* (60.1–70.2 Mb) identified in this study. Siddiqui et al. [26] reported 25 marker–trait associations (MTAs) for wheat seedling root traits in 200 diverse cultivars which were located on chromosomes 1A, 1B, 2A, 2B, 3A, 3B, 4B, 5A, 5D, 7A, and 7B. The locus on chromosome 3A (20.0–55.0 Mb) was similar to *QRT.gau-3AS* (60.1–70.2 Mb) but still different. Furthermore, Zaman et al. [27] reported 323 SNPs distributed across 20 loci after using GWAS for root system traits in 161 accessions, and they were mainly distributed on chromosomes 2A, 2B, 5A, 5D, 6A, 7B, and 7D. The loci on chromosomes 4A (639.6 Mb), 6A (542.5 Mb), and 6D (151.2 Mb) differed from *QRL.gau-4AL* (731.4–732.5 Mb), *QRV.gau-6AS* (31.0–45.3 Mb), and *QRT.gau-6DL* (462.6–468.3 Mb) identified in this study, respectively.

Using the Yangmai 16/Zhongmai895 DH population, Yang et al. [28] identified 13 QTLs for seedling traits on chromosomes 2B, 3B, 4A, 4D, and 7D. Among these, the QTL on chromosome 4A (18.0–40.5 Mb) differed from *QRL.gau-4AL* (4A: 731.4–732.5 Mb) identified in this study, whereas the loci on 3A (45.8–75.2 Mb) partially overlapped with *QRT.gau-3AS* (60.0–70.0 Mb). Liu et al. [29] reported 19 QTLs for seedling root traits, primarily distributed on chromosomes 1A, 2B, 2D, 3A, 3B, 3D, 5A, and 5D. The loci on chromosomes 2D (425.6 Mb) and 3A (556.9 Mb) differed from *QRA.gau-2DL* (439.6–446.0 Mb), *QRW.gau-2DL* (623.3–629.2 Mb), and *QRT.gau-3AS* (60.1–70.2 Mb). Alemu et al. [22] identified 38 QTLs for seedling root traits from 192 wheat varieties using association mapping. The loci on chromosome 4A (17.0 Mb) differed from *QRL.gau-4AL* (731.4–732.5 Mb) identified in this study. Although several studies on wheat seedling root traits have used traditional SSR and DArT markers [16,23], meaningful comparisons cannot be made based on the existing consensus map.

Genes affecting plant height (PH) and vernalization can also influence the wheat root system establishment. Several PH genes have been reported on chromosomes 2A (*Rht7* and *cqTN-2D.2*) [30,31], 2D (*Rht8*, *QPht/Sl.cau-2D.1*, *QPht/Sl.cau-2D.2*, and *qRht.2D*) [32,33,34], 3A (*Rht27* and *qRht.3A*) [35], and 6A (*QPh.cas-6A*, *Rht14*, and *Rht16*) [36,37,38]. Except for *QPh.cas-6A* (29.8 Mb), which is close to *QRV.gau-6AS* (31.0–45.3 Mb), no overlap was identified between the loci identified in this study and the reported PH genes. Additionally, no overlap was found among the vernalization genes (*VRN1*, *VRN2*, *VRN3*, and *VRN4*), the photoperiod gene *Ppd-D1*, and the root system loci identified in this study. Therefore, *QRL.gau-1DS*, *QRL.gau-1DL*, *QRA.gau-1D*, *QRA.gau-2DL*, *QRW.gau-2DL*, *QRW.gau-2AS*, and *QRT.gau-6DL* may be novel.

### 3.2. Candidate Genes for Wheat Seedling Root Traits

Plant hormones are the key internal factors that influence root system development. The primary regulatory hormones are IAA [39,40], CTK, and ETH [41]. IAA affects the entire root system development. At certain concentrations, IAA can inhibit root growth, promote cell differentiation, enhance cell division in meristematic tissues, and facilitate the formation of lateral and adventitious roots [41,42]. CTK can inhibit the transition of root primordium cells from the G2 phase to the M phase, affecting the development of adventitious roots.

The over-expression of CTK oxidase genes leads to a decrease in CTK levels and an increase in the number of adventitious roots [41,43]. In rice, the regulation of root system development by CTK and IAA is antagonistic; IAA promotes the formation of lateral and adventitious roots while CTK inhibits it. The synthesis and accumulation of ETH induce the growth of adventitious root primordia and the death of outer epidermal cells, thereby facilitating the formation of adventitious roots. The coordinated regulatory model involving CTK, IAA, and ETH suggests that IAA serves as the primary signaling molecule for adventitious root production [40,44]. IAA is transported polarly to the root tip through the pericycle, and ETH diffuses into adjacent tissues, regulating IAA and CTK transport in the target area to control the division of pericycle meristematic cells. Additionally, ETH influences lateral root formation by regulating the activity of the AUX1 protein and IAA transport. In addition, abscisic acid (ABA), gibberellin (GA), jasmonic acid (JA), and salicylic acid (SA) also participate in the regulation of adventitious root formation [40,45].

Seven genes were selected as high-confidence candidate genes and initially validated by qRT-PCR. The development of lateral roots in cereals involves the following three stages: organ initiation, cortex growth, and epidermis emergence [5], all of which are regulated by various plant hormones [15,46]. *TraesCS1D01G216500* of *QRA.gau-1D* encodes an IAA canalization protein, and *TraesCS2D01G548900* of *QRW.gau-2DL* encodes an IAA response factor (ARF). *TraesCS6D01G383600* of *QRT.gau-6DL* encodes an ethylene-regulated nuclear protein. *TraesCS1D01G336900* of *QRL.gau-1DL* encodes gibberellin 2-oxidase, which is essential in the catabolic pathway of gibberellins through 2β-hydroxylation [43] and controls semi-dwarfism, tillering, and root development [39]. Plant hormones, including IAA, ET, CKs, GA, ABA, BRs, and JA, directly influence root system development through their interactions with each other [47,48]. Of these, IAA serves as a basic signaling molecule, interacting with ETH, GA, and ABA and influencing seedling root growth [41,43,48].

The candidate gene *TraesCS1D01G018200* of *QRL.gau-1DS* encodes a zinc finger family (C2H2) protein. C2H2 family proteins are involved in primary root growth and shoot development. Chen et al. [49] reported that the over-expression of *TaZAT8-5B* (C2H2) enhances drought tolerance and root growth in *Arabidopsis thaliana*. *TraesCS3A01G101400* of *QRT.gau-3AS* encodes an E3 ubiquitin protein ligase, which is responsible for recognizing specific substrates and transferring ubiquitin to them, playing a crucial role in the entire process of plant growth [3]. *TraesCS2A01G220500* of *QRW.gau-2AS* and *TraesCS2D01G344600* of *QRA.gau-2DL* both encode leucine-rich repeat receptor protein kinases (LRR-RPKs). LRR-RPKs regulate seed germination by activating ABA-responsive genes in rice [50]. *TraesCS6A01G070300* of *QRV.gau-6AS* encodes an F-box family protein, which plays a significant role in signal transduction, cell differentiation, and stress tolerance in plants. However, no significant differences were observed between the parents of *TraesCS2A01G220500* and *TraesCS3A01G101400*.

### 3.3. Application of the KASP Markers for Wheat MAS Breeding

Modern wheat breeding primarily focuses on above-ground traits, including agronomic traits such as plant height and growth duration, yield such as thousand kernel weight and grain size, and resistance to diseases such as stripe rust and leaf rust. However, underground traits such as root system architecture are also closely linked to high and stable wheat yields. Although traditional breeding has made some progress in improving wheat seedling root system traits, the selection process remains lengthy and less efficient due to the challenges and labor-intensive nature of field measurements for underground traits. The previous studies have shown that seedling root development is critical for early wheat growth and significantly associated with high yields and stability. KASP markers have been widely adopted for detecting genetic variations in wheat, enabling high-throughput genotyping, which facilitates the efficient identification and selection of desirable traits in wheat breeding efforts [10]. KASP markers have been extensively applied to enhance the yield, disease resistance, and quality traits in wheat. In this study, we successfully developed five KASP markers; *KASP-RL-1DL* for RL, *KASP-RA-1D* and *KASP-RA-2DL* for RA, and *KASP-RW-2AS* and *KASP-RW-2DL* for RW were developed and validated successfully in 149 varieties, demonstrating their effectiveness as valuable tools in MAS breeding programs. In addition, based on the KASP markers we have developed and the root-related phenotypic data, we selected five varieties with excellent root phenotypes and favorable alleles as references for use in the breeding combination formulation for wheat stress tolerance (Appendix A).

## 4. Materials and Methods

### 4.1. Plant Materials

We evaluated seedling root traits using 243 F_2:6_ RILs derived from the cross between Wp-072 and Wp-119. W-072 is an advanced breeding line developed by Gansu Agricultural University, known for its tolerance to abiotic stress, high yield, and well-developed root systems. In contrast, Wp-119 has poor stress tolerance and weaker root development. A total of 40 mature seeds for each accession were sown in a plastic tray with nutrient soil (Pindstrup, Arhus, Denmark) at a depth of 13 cm, then the plastic tray was placed in a pallet with nutrient soil at a depth of 10 cm. The whole devices were then kept in a greenhouse (25 °C/65% RH and 14 h light/10 h dark) for about 21 days (tillering stage). Approximately 30 plants growing consistently for each accession were selected and surface-sterilized with 15% H_2_O_2_ for 15 min. Another 10 plants were collected for RNA extraction. Additionally, a diverse panel of 149 varieties, mainly from China, was used to validate the effectiveness of the developed KASP markers.

### 4.2. Phenotype Evaluation

First, the wheat roots were carefully washed with flowing deionized water, then arranged in an orderly manner in glass Petri dishes. Next, images were captured using an Expression 11000XL high-resolution scanner (Epson, Nagano-ken, Japan) and imported into the WinRHIZO root analysis system (LA6400XL). Default parameters were selected for both steps. Subsequently, a standard analytical balance was used for phenotypic assessment. The RL, RA, RV, and RT were recorded and statistically analyzed. After phenotyping with the WinRHIZO root analysis system, fresh root samples were collected and placed into kraft paper bags labeled with the corresponding identifiers, dried at 105 °C for 30 min, and then further dried at 80 °C for 24 h. The RW was recorded and statistically analyzed. In total, data were collected from 30 uniformly selected plants, and the mean values for all seedling root traits were taken as the final phenotypic values. Basic statistical analyses and frequency distributions were performed using SAS v9.3 (http://www.sas.com, accessed on 19 January 2025) and Excel 2021.

### 4.3. Linkage Map Construction

Using the wheat 90K SNP chip, the RIL population and parental lines were genotyped by CapitalBio, Beijing, China. After obtaining accurate genotype data, the markers were subjected to quality control based on the following criteria: (1) markers with no differences between the parents were filtered out; (2) heterozygous genotypes were treated as missing; (3) markers with a missing rate greater than 20% were filtered out; (4) markers with significant segregation distortion were excluded; and (5) low-quality SNP markers (missing rate < 0.2) were removed.

The high-quality SNP markers remaining after quality control were used to construct the genetic linkage map, following these steps: (1) Redundant marker filtering: Using the BIN function of IciMapping v4.1 [50] (http://www.isbreeding.net, accessed on 19 January 2025), the SNP markers were optimized by placing markers with identical genotypic information within the RIL population into a bin, treating them as a single genetic locus, deleting redundant markers, and selecting the marker with the lowest missing rate as the skeleton marker for mapping. (2) Marker grouping: The genotypes of the skeleton markers were imported into Joinmap v4.0 (Stam, 1993; http://www.kyazma.com, accessed on 19 January 2025) for grouping, with the LOD value ranging from 3 to 20. (3) Linkage map construction: Genotypes of markers within the same linkage group were imported into Joinmap v4.0, with the genetic distances calculated based on the Kosambi mapping function for linkage map construction. (4) Chromosome assignment: Based on the BLAST alignment results of the flanking sequences of the SNP markers with the latest released IWGSC v1.0 reference genome (http://www.wheatgenome.org/, accessed on 19 January 2025), the chromosomal positions of the markers were determined, and the linkage groups were anchored to the corresponding chromosomes.

### 4.4. Linkage Mapping and KASP Marker Development

Based on the genetic map of the Wp-072/Wp-119 RIL population, QTL mapping for five seedling root traits was performed using the Composite Interval Mapping (CIM) method in ICIMapping V4.1 [50]. The scanning step length was set to 1.0 cM, and after 2000 permutation runs at *p* ≤ 0.01, the LOD thresholds ranged from 1.8 to 2.7. To ensure the accuracy of the results, the LOD threshold was set to 2.7. The physical positions of the QTLs were derived from the IWGSC v1.0 reference genome. QTLs with higher PVEs values were used to develop KASP markers for subsequent application in wheat MAS breeding.

KASP primers consist of two competitive primers and one common primer [10]. Primer design was performed using PolyMarker (http://polymarker.tgac.ac.uk/, accessed on 19 January 2025). The PCR mix was prepared as follows: 40 μL of common primer (100 μM), 16 μL of each competitive primer (100 μM), and 60 μL of ddH_2_O. The reaction system included 2.5 μL of 2 × KASP master mix (LGC, Biosearch Technologies, Hoddesdon, UK), 0.07 μL of KASP primer premix, and 2.5 μL of template DNA (50 ng/μL). The PCR program consisted of pre-denaturation at 95 °C for 5 min, followed by 10 cycles of touchdown program (denaturation at 94 °C for 20 s, annealing at 65 °C for 25 s, decreasing by 0.8 °C per cycle), then 30 cycles (denaturation at 95 °C for 20 s, annealing at 57 °C for 60 s) and, finally, the PCR products were stored at 4 °C. After PCR amplification, fluorescence signals were detected using the PHERAstar Plus automatic focusing fluorescence multi-function microplate reader (BMG Labtech GmbH, Ortenberg, Germany). Genotyping was performed using the KlusterCaller software v4.1.2 (BMG Labtech GmbH, Ortenberg, Germany).

### 4.5. Identification of Candidate Genes for Wheat Seedling Root Traits

To identify the candidate genes for wheat seedling root traits, genes located within the linkage disequilibrium (LD) block region around the peak SNP (±5.0 Mb) of each QTL according to the IWGSC v1.0 were selected. To further validate the candidate genes identified based on QTL mapping and annotation, qRT-PCR was used to test the expression differences of candidate genes.

Twenty days after seeding (tillering stage), the root samples of Wp-072 and Wp-119 were collected for RNA extraction using the Trizol method. Subsequently, cDNA synthesis was performed with the HiScript II 1st Strand cDNA Synthesis Kit (Vazyme, Nanjing, China). Primers were designed using Primer Premier V5.0. The PCR reaction system was 20 μL, including 2 μL of cDNA (50 ng/μL), 10 μL of ChamQ Universal SYBR qPCR Master Mix, and 0.4 μL of each primer (10 μM). qRT-PCR was conducted on the ABI StepOne Plus Real-Time PCR System, and the gene expression levels were analyzed using the 2^−ΔΔCT^ method. All qRT-PCR assays for the candidate genes were designed with two biological replicates and three technical replicates. *TaActin1* was used as the reference gene.

## 5. Conclusions

In this study, a total of 10 loci for wheat seedling root system traits were identified in the Wp-072/Wp-119 RIL population. Among these, *QRL.gau-1DS*, *QRL.gau-1DL*, *QRA.gau-1D*, *QRA.gau-2DL*, *QRW.gau-2DL*, *QRW.gau-2AS*, and *QRT.gau-6DL* were novel. Additionally, *Kasp_4A_RL* for root length, *KASP-RA-1D* and *KASP-RA-2DL* for root angle, and *KASP-RW-2AS* and *KASP-RW-2DL* for root width were validated and can be applied in wheat MAS breeding. Finally, seven candidate genes were selected and validated by qRT-PCR. This study provides new loci and candidate genes, the available KASP markers, and novel varieties for optimizing wheat root system architecture.

## Figures and Tables

**Figure 1 plants-14-00490-f001:**
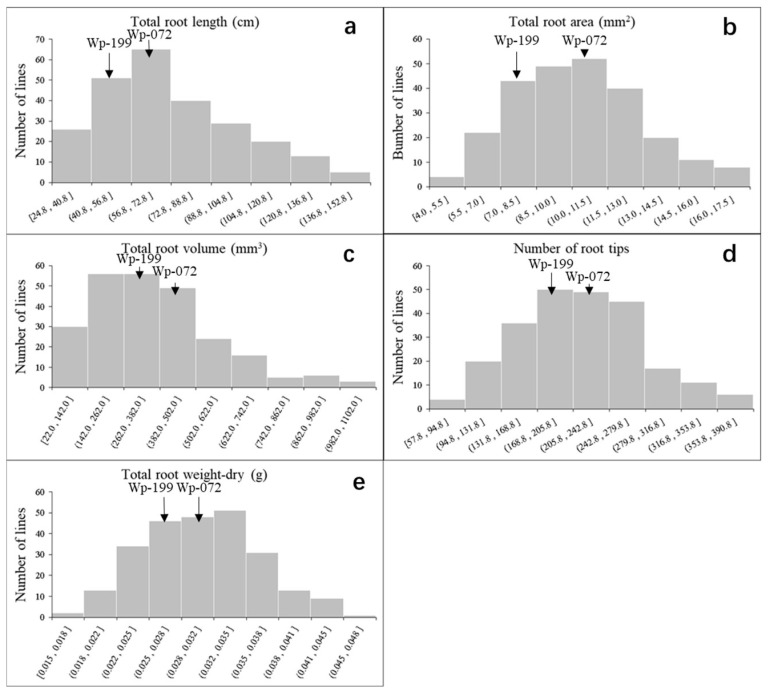
Frequency distributions for wheat seedling root traits in the Wp-072/Wp-119 RIL population. (**a**) RL, total root length (cm); (**b**) RA, total root surface area (mm^2^); (**c**) RV, total root volume (mm^3^); (**d**) RT, number of root tips; (**e**) RW, total root dry weight (g).

**Figure 2 plants-14-00490-f002:**
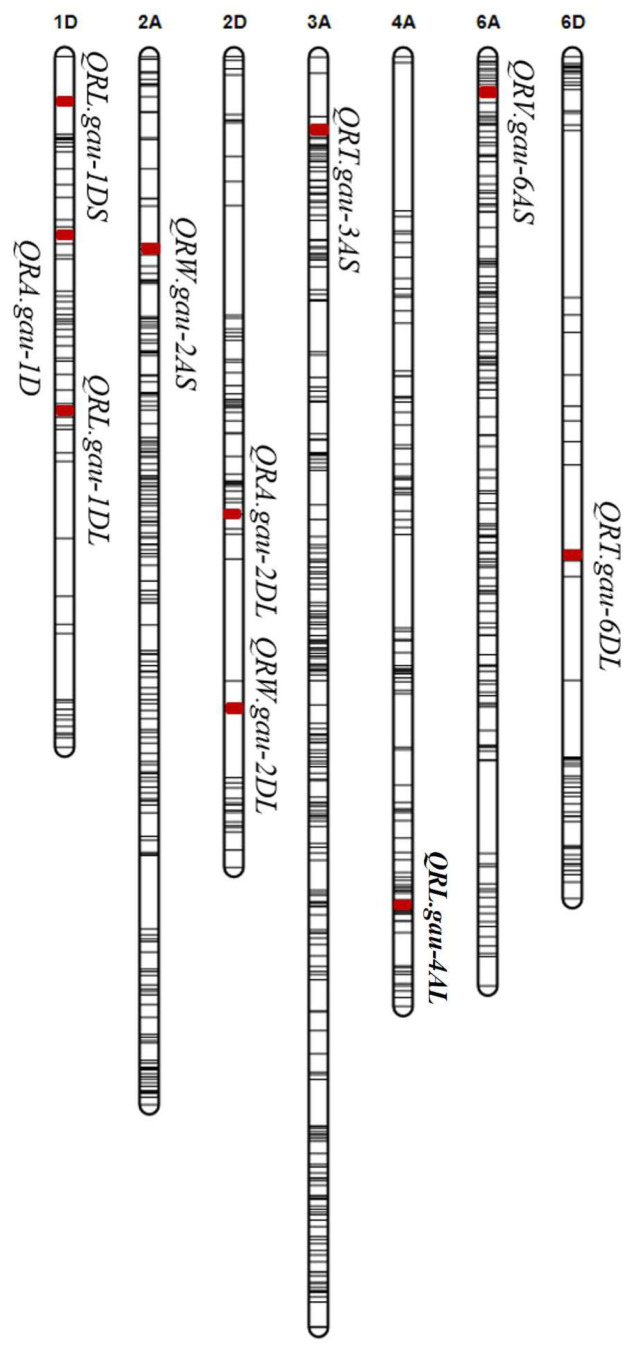
The identified QTLs for wheat seedling root traits in the Wp-072 × Wp-119 RIL population. The red filled area represents the position of QTL.

**Figure 3 plants-14-00490-f003:**
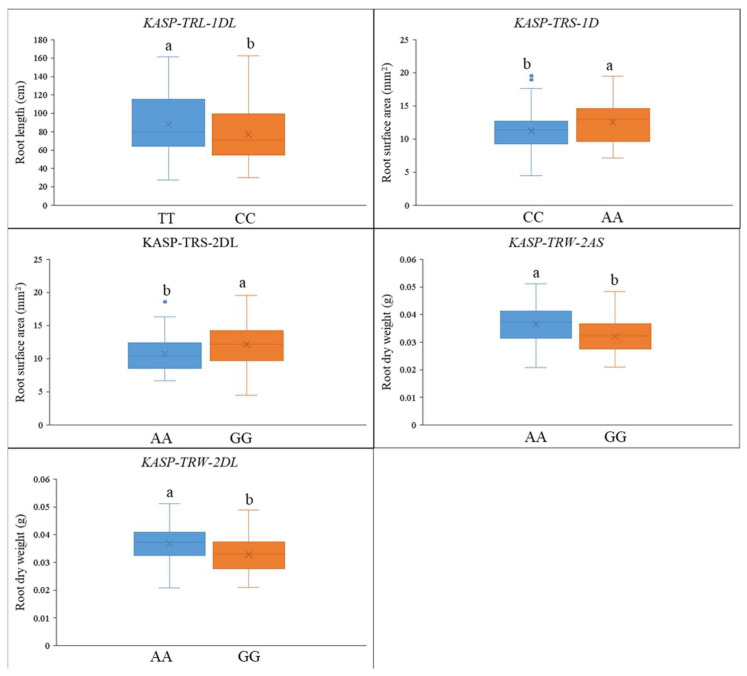
Validating the efficiency of KASP markers for root traits in the natural population. Lowercase letters a and b indicate significant differences at *p* = 0.05 level; “AA”, “CC”, and “GG” refer to the different base pairs; whiskers show the standard deviation (SD).

**Figure 4 plants-14-00490-f004:**
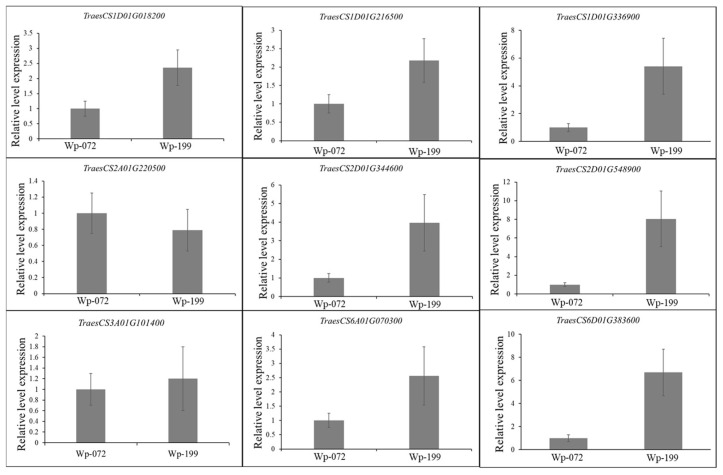
qRT-PCR of the nine candidate genes identified in Wp-072/Wp-119 RIL population.

**Table 1 plants-14-00490-t001:** QTL for wheat seedling root traits in Wp-072/Wp-119 RIL population.

QTL	Genetic Interval	Physical Position (Mb)	LOD	PVE	Add ^a^	Favorable Allele	Unfavorable Allele
*QRL.gau-1DS*	*Excalibur_c17152_454~* *RAC875_c1471_566*	7.1~7.9	2.7	5.2	−6.1	75.8 cm	72.4 cm
*QRL.gau-1DL*	*RAC875_c103613_441~* *Excalibur_c1236_840*	424.7~431.8	4.2	8.4	−9.2	76.0 cm	72.7 cm
*QRL.gau-4AL*	*BS00059454_51~* *Kukri_c19883_816*	731.4~732.5	2.7	5.4	−6.1	75.8 cm	72.3 cm
*QRA.gau-1D*	*Kukri_c18350_151~* *wsnp_Ex_rep_c70574_69491038*	294.5~302.8	4.1	8.2	−0.7	10.6 mm^2^	9.6 mm^2^
*QRA.gau-2DL*	*BobWhite_c6770_617~* *BS00026262_51*	439.6~446.0	4.3	8.4	0.7	10.7 mm^2^	9.8 mm^2^
*QRV.gau-6AS*	*CAP11_c4727_205~* *Tdurum_contig11539_81*	31.0~45.3	3.7	5.1	−58.2	389.2 mm^3^	325.6 mm^3^
*QRW.gau-2AS*	*BS00039489_51~* *Kukri_c15325_1360*	203.0~208.5	4.7	8.2	−0.0186	0.032 g	0.027 g
*QRW.gau-2DL*	*RAC875_c66820_684~* *Ku_c22718_1072*	623.3~629.2	4.2	8.1	0.0182	0.032 g	0.027 g
*QRT.gau-3AS*	*RAC875_c55313_89~* *Kukri_c64788_552*	60.1~70.2	2.6	4.5	−6.0	224.6	196.9
*QRT.gau-6DL*	*Ex_c31468_763* *tplb0055d02_624*	462.6~468.3	3.8	7.2	−7.5	218.7	190.2

^a^ ‘−’ indicates the effects originating from Wp-072.

**Table 2 plants-14-00490-t002:** The primers of the developed KASP markers for wheat seedling root traits.

KASP Marker	QTL	Primer	Sequence
*KASP-RL-1DL*	*QRL.gau-1DL*	FAM	AGGTGGGTTCTTCAAAGGAAT
		HEX	AGGTGGGTTCTTCAAAGGAAC
		Common	GAGAATGCAAATGAATCCTCTGG
*KASP-RA-1D*	*QRA.gau-1D*	FAM	GCTGACGCATTTGAAAAAGATACA
		HEX	GCTGACGCATTTGAAAAAGATACG
		Common	CACATTTCCTGCACGGAGAA
*KASP-RA-2DL*	*QRA.gau-2DL*	FAM	TCCATGTCGTTTTATAACATTGACA
		HEX	TCCATGTCGTTTTATAACATTGACG
		Common	GTGAACACTAAGTTGTTTGTGGTTA
*KASP-RW-2AS*	*QRW.gau-2AS*	FAM	TGTTAGAATCTTACCTACCAGCATA
		HEX	TGTTAGAATCTTACCTACCAGCATG
		Common	TCCGAGGATGGGTATTTAACATG
*KASP-RW-2DL*	*QRW.gau-2DL*	FAM	ATGGTCGTCAACTCCATACAA
		HEX	ATGGTCGTCAACTCCATACAG
		Common	GCATCAGTTCAACAAGGCTG

**Table 3 plants-14-00490-t003:** Effects of developed KASP markers for wheat seedling root traits in the natural population.

Marker Name	QTL	Genotype a	Number of Lines	Phenotype	*p*-Value
*KASP-RL-1DL*	*QRL.gau-1DL*	**TT**	56	276.6 cm	0.048 *
		CC	77	230.5 cm	
*KASP-RA-1D*	*QRA.gau-1D*	**AA**	56	12.5 mm^2^	0.014 *
		CC	84	11.2 mm^2^	
*KASP-RA-2DL*	*QRA.gau-2DL*	AA	53	10.7 mm^2^	0.007 **
		**GG**	89	12.1 mm^2^	
*KASP-RW-2AS*	*QRW.gau-2AS*	**AA**	66	0.037 g	0.001 **
		GG	79	0.032 g	
*KASP-RW-2DL*	*QRW.gau-2DL*	**AA**	41	0.037 g	0.002 **
		GG	102	0.033 g	

Bold indicates the favorable allele; * Significant at *p* < 0.05; ** Significant at *p* < 0.01.

**Table 4 plants-14-00490-t004:** The candidate genes for wheat seedling root traits identified in the Wp-072/Wp-119 RIL population.

QTL	Candidate Gene	Chr.	Start (bp)	End (bp)	Annotation
*QRL.gau-1DS*	*TraesCS1D01G018200*	1D	7,913,129	7,917,136	Zinc finger family protein
*QRA.gau-1D*	*TraesCS1D01G216500*	1D	302,349,108	302,352,162	Auxin canalization protein
*QRL.gau-1DL*	*TraesCS1D01G336900*	1D	426,869,492	426,870,633	Gibberellin 2-oxidase
*QRW.gau-2AS*	*TraesCS2A01G220500*	2A	208,242,970	208,243,194	Leucine-rich repeat receptor-like protein kinase
*QRA.gau-2DL*	*TraesCS2D01G344600*	2D	440,633,947	440,634,603	Leucine-rich repeat receptor-like protein kinase
*QRW.gau-2DL*	*TraesCS2D01G548900*	2D	624,361,783	624,369,474	Auxin response factor
*QRT.gau-3AS*	*TraesCS3A01G101400*	3A	65,952,815	65,953,444	E3 ubiquitin protein ligase
*QRV.gau-6AS*	*TraesCS6A01G070300*	6A	38,453,855	38,454,175	F-box family protein
*QRT.gau-6DL*	*TraesCS6D01G383600*	6D	462,483,712	462,487,539	Ethylene receptor

## Data Availability

All the data generated or analyzed during this study are included in this published article.

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
