# Peer review of "Genetic Basis of Seedling Root Traits in Common Wheat (*Triticum aestivum* L.) Identified by Genome-Wide Linkage Mapping"

_plants, 2025, doi:10.3390/plants14030490_

Round 1

Reviewer 1 Report

Comments and Suggestions for Authors

Review of the manuscript titled: Elucidating the genetic basis of root system related traits in common wheat (Triticum aestivum L.) by genome-wide linkage mapping.

The study is devoted to an important subject of identification of molecular markers for root traits. The material and methods are relevant for the study objective. The obtained results are valid, well analyzed, presented and interpreted. The study strength is also in validation of the results on additional panel. This is certainly important contribution to the knowledge and deserves publication.

The main challenge of the paper is the use of seedling traits only while referring to drought and abiotic stresses. There is a large school of though among breeders and physiologists who doubt the application of laboratory and especially seedlings traits in a real field situation. For this reason the authors may like to find and use references specifically mentioning the effects of seedling root traits in the improving drought or other stress tolerance in wheat. There are also several comments below.

1.      The title has a few words which are not needed. One option: Genetic basis of root traits in common wheat (Triticum aestivum L.) identified by genome-wide linkage mapping. Can be also “seedling root traits”.

2.      The authors refer to “root system related traits” but these are simply root system traits or root traits.

3.      The sentence on line 44-47 is too long and not clear.

4.      Section 2.1. The root traits values for parents need to be presented and ideally added to Figure 1.

5.      Section 2.3. Very long paragraph can be broken into a few according to traits.

6.      Table 1. Root trait for each marker has to be added.

7.      Tables A1, A2 or S1, S2.

8.      Section 2.4. It is difficult to understand which description refers to RIL population and which one to diverse panel. Authors also refer to natural population. Perhaps, one table will be justified showing the effects of the same markers on RILs and on validation panel.

9.      It would be good to make an additional table in the body of the paper or in the supplement showing 3-5 best varieties from validation panel for each root trait with their phenotypic expression and presence/absence of the respective alleles.

10.  English requires careful reading and editing to correct numerous mistakes. Line 14: “…related to root system related…”. Line 15: “…breeding high and stable wheat varieties…” – how high for plant height or for yield.

Comments on the Quality of English Language

In the review.

Author Response

The study is devoted to an important subject of identification of molecular markers for root traits. The material and methods are relevant for the study objective. The obtained results are valid, well analyzed, presented and interpreted. The study strength is also in validation of the results on additional panel. This is certainly important contribution to the knowledge and deserves publication.

The main challenge of the paper is the use of seedling traits only while referring to drought and abiotic stresses. There is a large school of though among breeders and physiologists who doubt the application of laboratory and especially seedlings traits in a real field situation. For this reason, the authors may like to find and use references specifically mentioning the effects of seedling root traits in the improving drought or other stress tolerance in wheat.

Response: Many thanks for your valuable suggestions. As you mentioned, our research is primarily conducted in the laboratory. The overwintering and drought resistance of winter wheat are significantly influenced by the root system development during the seedling stage. A well-developed and robust root system is fundamental for the successful overwintering of wheat. Furthermore, numerous studies have shown that a strong root system during the seedling stage is a prerequisite for high and stable yields in the later stages of wheat growth. Therefore, investigating the root system at the seedling stage holds considerable significance for achieving high yields in field conditions. In our preliminary experiments simulating wheat root growth, we observed that the root development of wheat seedlings was superior in nutrient soil compared to field soil and nutrient solution cultures. The nutrient soil environment most effectively demonstrates the developmental status of the wheat root system, facilitates easier cleaning, reduces loss, and yields more accurate phenotypic data. Based on these findings, we believe that this research also has substantial reference value for field production of wheat and can aid in wheat production and breeding. We greatly appreciate your reminder, as this aspect was not initially reflected in the Introduction section, and we have since supplemented it. Your suggestions have enhanced the readability of our article. Thank you once again.

  1. The title has a few words which are not needed. One option: Genetic basis of root traits in common wheat (Triticum aestivum L.) identified by genome-wide linkage mapping. Can be also “seedling root traits”.

Response: Many thanks for your valuable suggestions. We have revised it to “Genetic basis of seedling root traits in common wheat (Triticum aestivum L.) identified by genome-wide linkage mapping” in the update version.

  1. The authors refer to “root system related traits” but these are simply root system traits or root traits.

Response: Thank you for your kindly reminder. The term “root system related traits” has been replaced with “seeding root traits” throughout the manuscript for clarity and conciseness.

  1. The sentence on line 44-47 is too long and not clear.

Response: Many thanks for you kindly reminder. We will modify the particularly long sentence as follows: During the process of improving wheat landraces into cultivars, optimizing the root system has played a pivotal role. This optimization has not only significantly boosted yield and stress tolerance but has also underscored the importance of favorable root architecture [6-8].

  1. Section 2.1. The root traits values for parents need to be presented and ideally added to Figure 1.

Response: Many thanks for your kindly reminder. Parental data (WP-072 and WP-119) for root traits have been added to Section 2.1 and Figure 1 in the revised version.

  1. Section 2.3. Very long paragraph can be broken into a few according to traits.

Response: Thank you very much for your assistance. This was indeed an issue, as the excessively long paragraph made it very difficult to understand. Following your advice, we have divided the paragraph into three sections, making it much easier to read. We are particularly grateful for your suggestions.

  1. Table 1. Root trait for each marker has to be added.

Response: Thank you very much for your suggestions, which are very insightful. Since each QTL is determined by the left and right flanking markers, we did not include the phenotypic information for individual markers. Instead, we have added the phenotypic information corresponding to each QTL locus in Table 1. Your suggestion has made the results clearer and more useful for breeders. We are particularly grateful to you.

  1. Tables A1, A2 or S1, S2.

Response: Thank you very much for your reminder. We apologize for the mistake. The correct references are Tables S1 and S2. We misunderstood the Plants-Basel formatting requirements. In the new version, we have made the necessary corrections.

  1. Section 2.4. It is difficult to understand which description refers to RIL population and which one to diverse panel. Authors also refer to natural population. Perhaps, one table will be justified showing the effects of the same markers on RILs and on validation panel.

Response: We greatly appreciate your reminder. Our initial presentation lacked clarity and organization. We developed corresponding KASP markers based on the SNP markers located on both sides and within the identified QTL regions. Subsequently, we validated the reliability of these markers in the RIL population by checking the consistency between the KASP genotyping results and the 90K chip genotyping data. The KASP markers that passed this consistency validation were then used to assess their applicability and association with traits in the natural population. Therefore, the first paragraph of Section 2.4 discusses the RIL population, while the second and third paragraphs focus on the natural population. In the revised version, we retained the table showing the effects of KASP markers in the natural population and further clarified the work related to both the RIL and natural populations within the text. We believe these changes will enhance the logical flow and readability of the content.

  1. It would be good to make an additional table in the body of the paper or in the supplement showing 3-5 best varieties from validation panel for each root trait with their phenotypic expression and presence/absence of the respective alleles.

Response: Thank you for your suggestion. We believe your advice is very reasonable. We have made the necessary additions by selecting five representative natural varieties for each key locus and included them in the supplementary Table S4. Please kindly review the updated information.

  1. English requires careful reading and editing to correct numerous mistakes. Line 14: “…related to root system related…”. Line 15: “…breeding high and stable wheat varieties…” – how high for plant height or for yield.

Response: We sincerely appreciate your reminder. Upon reviewing the entire manuscript, we identified numerous English language errors. In accordance with your suggestions, we have thoroughly revised the entire article and engaged a native speaker to refine the wording and grammar. Thank you once again for your valuable advice and assistance in enhancing the quality of this paper.

Reviewer 2 Report

Comments and Suggestions for Authors

The manuscript with the title “Elucidating the genetic basis of root system related traits in common wheat (Triticum aestivum L.) by genome-wide linkage mapping” reports the results of a study focused on possibilities to optimize wheat root system architecture. For this reason, authors studied root traits in some recombinant inbred lines.

Results – first paragraph presents results of correlation (r^2) and regression coefficients (R^2) citing Table A2. Please insert footnote to table A2 and mention what type of correlation is? Pearson?

Figure 1 – The figure components shall be numbered and explained in the figure caption. One of the figure components located upper right has the vertical axis named “Bumber of…”. Nothing relevant is said in the text about this figure.

Figure captions shall always mention the test displayed and what whiskers represent SD, SE, CI?

General remark – the main strong point of this manuscript is that it focuses on underground (root traits), in contrast with wheat breeding efforts focused on above-ground traits, as it has been approached until now. The work presents an interesting perspective that shall be explored further.

Best regards.

Author Response

The manuscript with the title “Elucidating the genetic basis of root system related traits in common wheat (Triticum aestivum L.) by genome-wide linkage mapping” reports the results of a study focused on possibilities to optimize wheat root system architecture. For this reason, authors studied root traits in some recombinant inbred lines.

1 Results – first paragraph presents results of correlation (r^2) and regression coefficients (R^2) citing Table A2. Please insert footnote to table A2 and mention what type of correlation is? Pearson?

Response: Thank you very much for your reminder. This is indeed the correlation coefficient, and we have added a footnote in Table A2 legend.

2 Figure 1 – The figure components shall be numbered and explained in the figure caption. One of the figure components located upper right has the vertical axis named “Bumber of…”. Nothing relevant is said in the text about this figure.

Response: Thank you very much for your suggestions. We have now numbered each figure. Figure 1 represents the distribution of each trait in the RIL population, where the vertical axis is the number of lines, and the horizontal axis is the corresponding root trait value. We aim to demonstrate through this figure that the root-related traits in the RIL population exhibit a normal distribution. We have provided an explanation in Section 2.1. We especially appreciate your reminder; after making the revisions according to your suggestions, Figure 1 is much clearer and more informative.

3 Figure captions shall always mention the test displayed and what whiskers represent SD, SE, CI?

Response: Thank you very much for your reminder. We have added the caption for Figure 2. We used standard deviation (SD), which has also been noted in the caption. Thank you again for your help, which has made our article more standardized.

4 General remark – the main strong point of this manuscript is that it focuses on underground (root traits), in contrast with wheat breeding efforts focused on above-ground traits, as it has been approached until now. The work presents an interesting perspective that shall be explored further.

Response: Thank you very much for your positive feedback. As you mentioned, we focused on the root traits of wheat, hoping to further improve wheat through its root system, enhancing its high yield, stability, and tolerance to adverse conditions. As you pointed out, our article indeed has many areas that need to be standardized. After making revisions according to your suggestions, we look forward to presenting our results more clearly, providing a reference for high-yield and stable wheat breeding.

Reviewer 3 Report

Comments and Suggestions for Authors

The manuscript is prepared on an interesting and very current topic. Before its publication, I recommend modifications and additions, which I specify in my review.   

Introduction - in the first paragraph, the authors present the necessity and importance of breeding for the root system in times of climate change. I recommend incorporating the findings from the current article that deals with this issue (DOI: 10.17221/57/2024-CJGPB) and would suitably supplement the references from 2017 and 2022. In the Introduction section, we are missing at least a small paragraph or the inclusion of the importance of plant hormones, e.g. by moving from the introductory part of the discussion, where the authors address it, but in the list of phytohormones they omitted an important group, i.e. strigolactones (SLs). I recommend e.g. publication DOI: 10.17221/88/2024-CJGPB. Especially if the fundamental result, which was also reflected in the abstract, is the definition of 7 markers in connection with phytohormonal regulation.  

Before commenting on the Results and the Discussion section, which I have already partially done as part of the evaluation of the Introduction section. I would like to review the Materials and Methods section. In section 4.1. the authors state the initial characteristics of the genotypes that were used for the derivation of RILs, but it would be appropriate to supplement this with a reference to the publication where these facts will be declared. In section 4.2, it would be good to indicate the developmental stage of the plants (standard and international scale) when the assessment of the root system was carried out. In section 4.4. (line 399) it would be appropriate to unify the designation of genotypes (actually there is a different format throughout the manuscript - please unify). Line 403 the volumes (cDNA/primers) are given, but the essential concentration information is missing. Must be added.   

Results - are well described, but it would be good to supplement the figures with self-explanatory legends, e.g. Figure 3 - what are the line segments of the box graph missing, and what are the letters, Figure 4 - what are the line segments? SE or SD?   

Discussion - you already indicated above the possibility of moving some information about phytohormones to the Introduction section and including SLs in the discussion. 

Supplementary Materials - Table S1 some values ​​are missing and it is not clear why. Perhaps it would be appropriate to insert  e.g. "n.e." - not evaluated. Table S2 - explanations for asterisks are missing in the legend.  

The manuscript is a significant contribution to the study of the root system of wheat and the possibility of its breeding. In its current form, I recommend the manuscript for publication after major revision and second review. Although many of my notes and comments are more formal in nature.

Author Response

The manuscript is prepared on an interesting and very current topic. Before its publication, I recommend modifications and additions, which I specify in my review.   

  1. Introduction - in the first paragraph, the authors present the necessity and importance of breeding for the root system in times of climate change. I recommend incorporating the findings from the current article that deals with this issue (DOI: 10.17221/57/2024-CJGPB) and would suitably supplement the references from 2017 and 2022. In the Introduction section, we are missing at least a small paragraph or the inclusion of the importance of plant hormones, e.g. by moving from the introductory part of the discussion, where the authors address it, but in the list of phytohormones they omitted an important group, i.e. strigolactones (SLs). I recommend e.g. publication DOI:10.17221/88/2024-CJGPB. Especially if the fundamental result, which was also reflected in the abstract, is the definition of 7 markers in connection with phytohormonal regulation.

Response: Thank you very much for your suggestions; they are very meaningful. In our initial conception, we placed the hormone section in the Introduction, but felt that the content was too extensive, so we moved it to the Discussion. We believe your suggestion is very reasonable, and we have made adjustments to reintegrate the relevant content into the Introduction. Additionally, the reference you provided (DOI: 10.17221/88/2024-CJGPB and DOI: 10.17221/88/2024-CJGPB) is particularly important, clearly and concisely demonstrating the relationship between the root system and plant hormones. We have cited this reference. We have also modified the abstract to highlight that 7 candidate genes are related to plant hormones.

  1. Before commenting on the Results and the Discussion section, which I have already partially done as part of the evaluation of the Introduction section. I would like to review the Materials and Methods section. In section 4.1. the authors state the initial characteristics of the genotypes that were used for the derivation of RILs, but it would be appropriate to supplement this with a reference to the publication where these facts will be declared.
    Response: Thank you very much for your suggestions. We apologize for the confusion caused by our incorrect statement. Both parents, Wp-072 and Wp-119, are advanced breeding lines obtained during our drought tolerance breeding program in the western part of China, and they have not undergone variety registration. We have accumulated root phenotype data for these two parents over multiple years and environments, but this data has not been published. We have revised the text at this point to avoid any misunderstanding by the readers. We greatly appreciate your valuable suggestions.
  2. In section 4.2, it would be good to indicate the developmental stage of the plants (standard and international scale) when the assessment of the root system was carried out.

Response: Thank you very much for your suggestions. We conducted the measurements 21 days after wheat sowing, during the tillering stage. We have made the necessary revisions in the new version. Thank you again for your feedback.

4 In section 4.4. (line 399) it would be appropriate to unify the designation of genotypes (actually there is a different format throughout the manuscript - please unify).

Response: Thank you very much for your suggestions. We have made the revisions according to your request. All instances of genotype names have been standardized.

  1. Line 403 the volumes (cDNA/primers) are given, but the essential concentration information is missing. Must be added.

Response: Thank you very much for your suggestions. This was our oversight, and we have added the concentration in the new version. The PCR reaction system was 20 μL, including 2 μL of cDNA (50 ng/μL), 10 μL of ChamQ Universal SYBR qPCR Master Mix, and 0.4 μL of each primer (10 μM).

  1. Results - are well described, but it would be good to supplement the figures with self-explanatory legends, e.g. Figure 3 - what are the line segments of the box graph missing, and what are the letters, Figure 4 - what are the line segments? SE or SD?

Response: Thank you very much for your suggestions. Indeed, our previous version was not written to a sufficiently high standard. In the new version, we have added tables and figure captions for each graph, which will better assist the readers in understanding the content. Specifically, we have included the meanings of the missing lines and letters in Figure 3, and we have noted that the standard deviation is used in Figure 4. We appreciate your efforts in helping to improve this paper once again.

  1. Discussion - you already indicated above the possibility of moving some information about phytohormones to the Introduction section and including SLs in the discussion. 

Response: Thank you very much for your suggestions. We fully agree with your opinions and have added the relevant content to the Introduction section, which will make the article more organized and clear.

  1. Supplementary Materials - Table S1 some values ​​are missing and it is not clear why. Perhaps it would be appropriate to insert e.g. "n.e." - not evaluated.

Response: Thank you very much for your reminder. We have noted in the footnote that the corresponding phenotypes for some strains are missing.

  1. Table S2 - explanations for asterisks are missing in the legend.

Response: We are particularly grateful for your suggestion. This was indeed an oversight on our part, and we have now added the meanings of the asterisks in the new table and supplemented them in the legend. Please review.

  1. The manuscript is a significant contribution to the study of the root system of wheat and the possibility of its breeding. In its current form, I recommend the manuscript for publication after major revision and second review. Although many of my notes and comments are more formal in nature.

Response: We are particularly grateful for your suggestions, which are very objective and comprehensive, and significantly contribute to improving the quality and readability of our article. In addition, all references and abbreviations were cross-checked for consistency. Typos were corrected. We look forward to this research providing a reference for the high-yield and stable-yield breeding of wheat. We also especially hope that with your assistance, the article can be published, offering more references for the high-yield, stable-yield, and root system optimization breeding of wheat. Thank you again for your great efforts in improving the quality of our article.

Round 2

Reviewer 3 Report

Comments and Suggestions for Authors

The authors accepted all my comments, which were suitably supplemented by comments. Just one small note about Figure 4, where the legend lacks an explanation as to whether the line segments indicate SD or SE. This can be removed during author (print) proofreading.